# Hydrogel-Based Formulations to Deliver Analgesic Drugs: A Scoping Review of Applications and Efficacy

**DOI:** 10.3390/biomedicines13102465

**Published:** 2025-10-10

**Authors:** Sveva Di Franco, Aniello Alfieri, Pasquale Sansone, Vincenzo Pota, Francesco Coppolino, Andrea Frangiosa, Vincenzo Maffei, Maria Caterina Pace, Maria Beatrice Passavanti, Marco Fiore

**Affiliations:** 1Department of Women, Child and General and Specialized Surgery, University of Campania “Luigi Vanvitelli”, 80138 Naples, Italy; pasquale.sansone@unicampania.it (P.S.); vincenzo.pota@unicampania.it (V.P.); francesco.coppolino@unicampania.it (F.C.); mariacaterina.pace@unicampania.it (M.C.P.); mariabeatrice.passavanti@unicampania.it (M.B.P.); 2Department of Anaesthesia and Intensive Care, P.O. Pellegrini, 80134 Naples, Italy; 3Department of Anaesthesia and Intensive Care, A.O.R.N. Antonio Cardarelli, 80131 Naples, Italy; andrea.frangiosa@gmail.com (A.F.); vincenzo.maffei@aocardarelli.it (V.M.)

**Keywords:** hydrogels, drug delivery systems, analgesics, pain management, scoping review

## Abstract

**Background/Objectives**: Hydrogels are highly hydrated, biocompatible polymer networks increasingly investigated as drug-delivery systems (DDS) for analgesics. Their ability to modulate local release, prolong drug residence time, and reduce systemic toxicity positions them as promising platforms in perioperative, chronic, and localized pain settings. This scoping review aimed to systematically map clinical applications, efficacy, and safety of hydrogel-based DDS for analgesics, while also documenting non-DDS uses where the matrix itself contributes to pain modulation through physical mechanisms. **Methods**: Following PRISMA-ScR guidance, PubMed, Embase, and Cochrane databases were searched without publication date restrictions. Only peer-reviewed clinical studies were included; preclinical studies and non-journal literature were excluded. Screening and selection were performed in duplicate. Data extracted included drug class, hydrogel technology, clinical setting, outcomes, and safety. Protocol was registered with Open Science Framework. **Results**: A total of 26 clinical studies evaluating hydrogel formulations as DDS for analgesics were included. Most were randomized controlled trials, spanning 1996–2024. Local anesthetics were the most frequent drug class, followed by opioids, corticosteroids, Non-Steroidal Anti-Inflammatory Drugs (NSAIDs), and neuromodulators. Application sites were predominantly topical/transdermal and perioperative/incisional. Across the DDS cohort, most of the studies reported improved analgesic outcomes, including reduced pain scores and lower rescue medication use; neutral or unclear results were rare. Safety reporting was limited, but tolerability was generally favorable. Additionally, 38 non-DDS studies demonstrated pain reduction through hydrogel-mediated cooling, lubrication, or barrier effects, particularly in burns, ocular surface disorders, and discogenic pain. **Conclusions**: Hydrogel-based DDS for analgesics show consistent clinical signals of benefit across diverse contexts, aligning with their mechanistic rationale. While current evidence supports their role as effective, well-tolerated platforms, translational gaps remain, particularly for hybrid nanotechnology systems and standardized safety reporting. Non-DDS applications confirm the intrinsic analgesic potential of hydrogel matrices, underscoring their relevance in multimodal pain management strategies.

## 1. Introduction

Hydrogels—highly hydrated, biocompatible, and tunable three-dimensional polymer networks—have emerged as versatile platforms for DDS. Their ability to encapsulate and release active pharmaceutical ingredients with controlled profiles, while simultaneously reducing undesired systemic exposure, is a key advantage. These properties stem from a delicate balance between their crosslinked structure, water content, and release mechanisms (e.g., diffusion, degradation, and swelling), which allows formulations to be tailored to specific therapeutic contexts. “Smart” or stimuli-responsive hydrogels (sensitive to pH, temperature, enzymes, light, or redox conditions) add a further layer of temporal and spatial precision to drug release. Furthermore, hybrid hydrogel–nanotechnology systems integrate nanoparticles or other nanostructures to enhance drug loading, stability, and targeting [1,2].

In the management of pain, interest in hydrogels is driven by the goal of enhancing the clinical efficacy of analgesics while minimizing systemic toxicity and exposure variability, particularly for hydrophobic molecules or those with a narrow therapeutic window. The use of natural and synthetic matrices—including injectable or in situ-forming variants—aims to prolong the local residence time of the drug, mitigate the initial “burst release,” and promote tissue targeting in settings where systemic administration is suboptimal. The recent literature indicates a growing interest in sustained-release approaches for analgesics (including local anesthetics) and in stimuli-responsive platforms that allow for finer control over release kinetics [3,4].

Given the broad technological scope (polymer origin, crosslinking methods, loading modalities, responsiveness) and the variety of relevant outcomes (e.g., targeting precision, reduction in dose-dependent adverse events, improved bioavailability, and delivery of hydrophobic drugs), a scoping review is an appropriate method to systematically map formulations, mechanisms, and applications in the analgesic domain, without being limited to a single class of molecules or setting. Moreover, the emergence of hybrid hydrogel–nanotechnology systems suggests new interdisciplinary research directions—from materials design to clinical pharmacokinetics—that warrant a structured reconnaissance [5,6]. Objective. Over the past 20 years, we have synthesized the evidence published in peer-reviewed journals on hydrogel formulations used for the delivery of analgesic drugs, with an emphasis on applications and efficacy. In this review, efficacy is operationalized as targeting precision, reduction in systemic toxicity, improved bioavailability, and the capacity to deliver hydrophobic drugs; we also highlight key gaps and advances in hybrid hydrogel–nanotechnology systems.

Beyond drug delivery, hydrogels are widely used as advanced dressings for wounds and burns, where they maintain a moist microenvironment, facilitate re-epithelialization, and can reduce pain (e.g., during dressing changes) through cooling, reduced adherence to the wound bed, and atraumatic removal. A clinical meta-analysis reported reductions in pain scores with hydrogels compared to controls in various types of skin lesions, including second-degree burns and traumatic wounds. Systematic reviews of superficial/partial-thickness burns and the recent wound management literature confirm that hydrogels can minimize pain on removal compared to more adherent dressings [7,8].

In the domain of tissue hydration and lubrication, hydrogels (e.g., cross-linked hyaluronates in ophthalmic formulations) act as surface lubricants, prolong precorneal retention, and improve signs and symptoms of dry eyes, including discomfort/pain from friction. In gynecology, moisturizing and bioadhesive gels based on hydrophilic polymers (hydrogel-like) have been shown in randomized trials to reduce dryness and dyspareunia, thereby modulating pain without delivering active analgesics. While these applications fall outside the primary scope of this review (focused on hydrogels for analgesic release), they are relevant as they demonstrate how the hydrogel matrix itself can contribute to pain modulation through physical mechanisms (lubrication, barrier, moisture, and atraumaticity) [9,10,11].

The main purpose of this scoping review is to systematically map and critically describe clinical evidence on hydrogel-based analgesic drug delivery systems—including drug classes, hydrogel technologies, administration routes, and reported outcomes—and to contrast these with non-DDS applications to provide a comprehensive overview for clinicians and researchers.

## 2. Materials and Methods

This Scoping Review Protocol was registered with Open Science Framework (https://osf.io/7yfnb/, accessed on 3 October 2025); it was designed and reported in accordance with the PRISMA-ScR (Preferred Reporting Items for Systematic reviews and Meta-Analyses extension for Scoping Reviews) checklist [12]. The PRISMA flow diagram, summarizing the study inclusion process, is illustrated in Figure 1.

Only articles published in peer-reviewed journals were included; gray literature was excluded, which is consistent with the objective of transparently and reproducibly mapping clinical evidence as intended for scoping reviews. PubMed/MEDLINE, Embase, and Cochrane databases were searched with no publication date restrictions. The complete search strings for each database (constructed by combining controlled vocabulary and keywords for hydrogel, drug delivery, and pain) are reported in in Table A1 (Appendix A).

Records retrieved from the databases were aggregated into a single library and deduplicated prior to screening.

Inclusion criteria adopted were as follows: clinical studies only (randomized clinical trials, RCTs, and non-randomized, prospective/retrospective, cohort, case–control, and case series/reports) on hydrogel formulations (natural or synthetic, including stimuli-responsive and hybrid systems with nanotechnologies) used as drug-delivery systems for analgesic drugs. Exclusion criteria were as follows: preclinical studies (in vitro/in vivo animal models), formulation-only or proof-of-concept works without clinical data, non-journal articles (proceedings, theses, preprints, and websites), and irrelevant studies (non-hydrogel or not related to analgesics).

For this scoping review, hydrogels were classified as DDS when the formulation explicitly contained and released an analgesic active pharmaceutical ingredient with the primary aim of delivering the drug locally. In contrast, studies in which the hydrogel acted without an active analgesic—modulating pain through physical or biophysical mechanisms (e.g., cooling, lubrication, and barrier effect)—were considered non-DDS and excluded from the primary synthesis but cataloged separately for transparency. This methodological choice allows readers to distinguish pharmacological versus non-pharmacological mechanisms and to appreciate the broader clinical landscape of hydrogel use in pain modulation. This distinction between DDS and non-DDS hydrogels is relevant for interpretation: the primary synthesis focuses on clinical outcomes attributable to drug delivery, while the non-DDS studies (cataloged separately) illustrate the intrinsic pain-modulating potential of the matrix itself. Presenting both groups clarifies the mechanism of action and contextualizes the scope of hydrogel applications in pain management.

The selection was performed in two phases: a first phase of title/abstract screening against the eligibility criteria, and a second phase of full-text assessment of potentially relevant articles. Screening was conducted independently by paired reviewers, with discrepancies resolved by consensus; a third reviewer was consulted in cases of persistent disagreement. Reasons for exclusion at the full-text stage were documented and summarized in the PRISMA-ScR flow diagram. For each included study, the following data were extracted: clinical context/pain indication; hydrogel type; analgesic molecule/class; administration route/target; study design; and main results. In line with scoping review recommendations, a formal risk of bias assessment was not planned.

As this work is a scoping review, no formal inferential statistics or meta-analysis were performed. Extracted data were summarized using descriptive statistics (counts, percentages, and qualitative thematic grouping) to map study characteristics, drug classes, hydrogel technologies, and reported outcomes.

## 3. Results

The search of PubMed/MEDLINE, Embase, and Cochrane identified a total of 838 records; after deduplication, 493 unique records remained. Title and abstract screening, followed by full-text assessment according to the predefined criteria, led to the inclusion of 87 clinical articles relevant to the hydrogel–pain theme. Within the included set, 26 studies specifically evaluated hydrogel formulations as DDS for analgesics and constitute the primary cohort for the thematic mapping of efficacy and safety. In parallel, the screening identified 38 clinical studies on hydrogel applications involving pain modulation not based on drug delivery (e.g., wound care, and hydration/lubrication): this evidence was cataloged separately and does not contribute to the primary DDS-oriented synthesis but is reported for methodological completeness and its potential clinical relevance. The PRISMA-ScR flow diagram summarizes the flow of records through the identification, screening, and inclusion phases, with documentation of reasons for exclusion at the full-text stage.

The screening led to the inclusion of 26 clinical studies evaluating hydrogel formulations as DDS for analgesics (primary cohort). The majority had a randomized design (18/26; 69.2%), followed by case series (2/26; 7.7%), pilot/open label/pharmacokinetic studies (4/26; 15.4%), and retrospective studies (2/26; 7.7%). Figure 2 illustrates the distribution of study designs among the 26 included clinical studies, showing a clear predominance of randomized trials over other designs. The timeframe of the studies ranged from 1996 to 2024 (median 2021).

By application site/route, the distribution shows a predominant use for topical/transdermal (10/26; 38.5%) and perioperative/incisional/wound (8/26; 30.8%) applications, with smaller contributions from oral/mucosal (2/26; 7.7%), rectal/anal (2/26; 7.7%), intra-articular (2/26; 7.7%), paranasal sinuses (1/26; 3.8%), and regional block (1/26; 3.8%). Regarding the pain context, studies were mainly perioperative (acute post-operative pain) (9/26; 34.6%), with the remaining indications being heterogeneous (chronic ulcers/wounds 4/26; 15.4%, knee osteoarthritis 2/26; 7.7%, and single occurrences in acute oral pain, burns, chronic neck pain, acute sinusitis, anal fissure, neuropathic pain, chronic low back pain, mucositis, pediatric pain, and toe pain). Figure 3 depicts the distribution of application sites across the included clinical studies.

Regarding drug class, local anesthetics were most prevalent (12/26; 46.2%; e.g., lidocaine, ropivacaine, and bupivacaine), followed by opioids (5/26; 19.2%; e.g., morphine, loperamide), intra-articular corticosteroids (2/26; 7.7%; triamcinolone in Hyaluronic Acid, HA-gel), NSAIDs (1/26; 3.8%; loxoprofen), topical neuromodulators (1/26; 3.8%; capsaicin), and nitric oxide (NO) donors for anal fissure (1/26; 3.8%). In four studies (15.4%), the nature of the active agent or the analgesic profile was not uniquely classified within the main categories but fell within the analgesic clinical scope.

As for hydrogel technology, various types were observed: thermo-responsive hydrogels (poloxamer/proprietary; 5/26; 19.2%), hyaluronates (2/26; 7.7%, including HA-steroid systems), muco-adhesives or bio-adhesives (2/26; 7.7%), amorphous wound gels (2/26; 7.7%), hydrophilic suppositories (1/26; 3.8%), and cellulosic gels (1/26; 3.8%). In approximately 13/26 studies (50.0%), the description of the material or platform did not allow for a more detailed classification; no included study explicitly reported a hybrid integration with nanotechnologies within the hydrogel matrix. Figure 4 illustrates the variety of hydrogel technologies employed across the included studies.

The clinical outcomes measured were consistent with the review’s scope: pain control, consumption of rescue analgesics, time to/duration of the effect, tolerability, and local or systemic adverse events. In some perioperative studies with local anesthetics, clinical pharmacokinetic proxies were reported. Comparators included placebo, standard of care, or non-hydrogel formulations; in non-controlled studies, interpretation was based on pre- and post-outcomes and feasibility.

In parallel, 38 non-DDS clinical studies were identified (excluded from the primary synthesis), in which the hydrogel modulated pain through physical mechanisms (advanced dressings, post-photorefractive keratectomy (PRK) bandage contact lenses (BCL), intradiscal implants, and tissue hydration/lubrication). In this group, ocular (≈24% of the excluded set), burns and wounds (≈20%), and discogenic/orthopedic pain contexts were frequent; these studies were designed as case series or randomized studies. This evidence is cataloged separately and does not feed into the efficacy mapping of DDS for analgesics, but illustrates the broad spectrum of hydrogel use in “drug-free” pain modulation.

In the primary cohort (26 clinical studies on hydrogels used as drug-delivery systems for analgesics), the most frequently reported clinical outcomes were pain intensity (Visual Analogue Scale, VAS/Numerical Rating Scale, NRS), consumption of rescue analgesics, time to pain relief, pain-free period, and tolerability/adverse events. Based on the summaries provided, 22 studies reported a relevant improvement in analgesic outcomes compared to control or baseline (22/26; 84.6%), with a limited number of neutral results (1/26; 3.8%) and a few cases that were not clearly determinable due to insufficient detail (3/26; 11.5%). Safety reports were found in a minority of studies, while tolerability was widely described as favorable.

### 3.1. Drug Classes

Local Anesthetics (12/26; 46.2%). Designs were mostly randomized; predominant routes of administration were perioperative, incisional, or wound injection (eight studies), with contributions from topical/transdermal (two), paranasal sinuses (one), and regional block (one). Outcomes included reduction in VAS/NRS, decrease in rescue medication use, and prolonged duration of analgesic effect; 10 of these 12 studies that analyzed local anesthetics in pain management reported great pain-solving effect, while 1 study was neutral and 1 was not definable.Opioids and Peripheral Opioid Receptor Modulators (5/26; 19.2%). To administer these drugs, there was a prevalence of topical and transdermal application (three), with evidence also for rectal/anal (one) and oral/mucosal (one) routes. The main endpoints were pain release and need for rescue medication; four of five studies reported improvement in pain management, and one was not definable.Intra-articular Corticosteroids (2/26; 7.7%). Intra-articular (i.e., knee infiltration) in an osteoarthritis context; both studies described pain relief and better clinical outcomes.NSAIDs (1/26; 3.8%). Topical/transdermal application guaranteed good pain control and outcomes.Topical Neuromodulator (i.e., capsaicin) (1/26; 3.8%). Topical/transdermal application of capsaicin improved pain relief if the pain was related to local symptoms.Other/Not Classified (4/26; 15.4%). Heterogeneous contexts (topical/transdermal, oral/mucosal, and perioperative/incisional/wound); three studies reported good pain management, and one was not definable.

Figure 5 shows the distribution of drug classes across the included clinical studies, with local anesthetics representing nearly half of all interventions.

### 3.2. Route and Site of Administration

Hydrogel application was predominantly topical and transdermal (10/26; 38.5%), followed by their use in perioperative, incisional, and wound settings (8/26; 30.8%), or by oral and mucosal route (2/26; 7.7%), rectal or anal (2/26; 7.7%), and intra-articular delivery (2/26; 7.7%). The application in the frontal sinus cavity was reported in one case (1/26; 3.8%), such as their usage in regional block anesthesia (1/26; 3.8%). The direction of effect was positive across all site categories (with single neutral or undefinable cases in perioperative/incisional/wound and topical/transdermal).

### 3.3. Hydrogel Technology

Thermo-responsive (poloxamer/proprietary; 5/26; 19.2%), hyaluronate (2/26), muco/bio-adhesive (2/26), amorphous wound gels (2/26), hydrophilic suppositories (1/26), and cellulosic (1/26) platforms were identified. In 11 studies of the total 26 (42.3%), the description of the matrix did not allow a precise classification (other/not specified). None of the included clinical studies reported hybrid hydrogel–nanotechnology systems in the examined formulation.

In Table 1, there is a short report that summarizes DDS hydrogels’ characteristics and settings of usage, while a comprehensive report is presented in Table A2 (Appendix B).

### 3.4. Non-DDS Clinical Studies (Pain Modulation Without Drug Release)

In the 38 non-DDS clinical studies (cataloged separately), the hydrogel modulated pain through physical mechanisms (protection, cooling, lubrication, atraumatic barrier, and adhesion), without acting as a drug vehicle. The prevalent areas were ocular (≈24%), burns or wounds (≈20%), and musculoskeletal and discal damage. These were mostly case series and randomized studies. A thematic summary follows.

#### 3.4.1. Ocular

Silicone hydrogel BCLs are used to reduce pain and promote re-epithelialization after PRK; comparative studies and clinical trials show benefits in pain and comfort, with differences also related to lens design, material, and curvature. In comparison of silicone-hydrogel BCLs, the authors reported improved pain management and patients’ comfort and epithelial stability; other trials evaluated fitting parameters comparing different silicone-hydrogels, and confirmed the reduction in post-PRK discomfort with good tolerability profiles. This evidence supports the advantage of using BCLs as a non-pharmacological analgesic intervention in the early stages post-corneal ablation [38,39].

#### 3.4.2. Burns and Wounds

In burn patients (especially pediatric), clinical trials have evaluated cooling and moisturizing hydrogel dressings to alleviate pain immediately after first aid and during initial dressing changes. A pediatric RCT showed a reduction in pain scores with a hydrogel compared to plastic film, while protocols and further studies have investigated the early analgesic effect of specific products. Recent reviews emphasize that cooling, water retention, and atraumatic adhesion contribute to pain reduction and better dressing management [40]. Overall, the evidence indicates that hydrogel dressings can be a non-pharmacological analgesic adjuvant in superficial/partial-thickness burns and some painful wounds, despite heterogeneity in settings and outcomes [40,41].

#### 3.4.3. Musculoskeletal and Discal

In discogenic pain, the use of intradiscal hydrogel implants/augmentations aims to restore hydration/cushioning of the nucleus pulposus with potential pain benefits. However, the clinical literature highlights safety signals: cases of hydrogel migration into the spinal canal with radicular pain or neurological complications requiring invasive management have been reported. Recent reviews underscore the insufficiency of controlled clinical data and the need for rigorous prospective evaluations before widespread clinical adoption. In this area, therefore, the “non-DDS” use appears promising but still immature, with a primary focus on safety [42,43].

#### 3.4.4. Other Contexts (Lubrication/Barrier)

At mucosal or cutaneous sites, bio-adhesive and muco-adhesive hydrogels or hydrophilic sheets and gels have been used as moisturizing and lubricating barriers, reducing pain from friction or dressing removal and improving the overall tolerability of local care. Clinical evidence is heterogeneous and often small-scale but converges on the idea that the hydrogel matrix, even without a drug, can help reduce procedural and incidental pain through physical effects (moisture, cooling, protection, atraumaticity) [44].

In Table 2, there is a report that summarizes the non-DDS hydrogels’ characteristics and the setting of usage.

## 4. Discussion

This scoping review clinically maps the use of hydrogels as drug-delivery platforms for analgesics (DDS), revealing a heterogeneous landscape in terms of drug classes, hydrogel types, administration sites, and routes, but with a predominantly favorable direction of effect on pain outcomes and the use of rescue analgesics.

Baseline pain severity varied widely across the included studies, from severe acute postoperative pain (e.g., thoracic, abdominal, or orthopedic surgery) to milder chronic or procedural pain (e.g., dry eye, topical applications). Most perioperative trials enrolled patients with moderate-to-severe pain and reported clearer benefits of hydrogel-based DDS compared with controls, particularly in reducing rescue analgesic use and prolonging the analgesic effect. In chronic or lower-intensity pain contexts, outcomes were more heterogeneous, with smaller effect sizes and occasional neutral results. This suggests that hydrogel-based DDS may be more consistently effective in high-intensity, acute pain settings, whereas efficacy in chronic or milder conditions appears variable and formulation dependent.

These findings are consistent with the engineering rationale of hydrogel-based DDS, the ability to modulate release, local residence time, and tissue concentrations, which is well-documented in the literature on design and mechanisms (e.g., mesh size, hydrogel–drug interactions, crosslinking, and responsiveness) and associated with clinical benefits when local tissue distribution is a key determinant of the outcome [45].

In perioperative/incisional contexts, thermo-hydrogels (e.g., poloxamer 407) that in situ have enabled the “single-shot” application of local anesthetics with outcomes non-inferior to continuous catheter techniques and with less procedural complexity, suggesting a possible operational advantage and a limited systemic toxicity profile. In an RCT in minimally invasive thoracic surgery, local injection of ropivacaine in poloxamer 407 was non-inferior to a continuous paravertebral block for postoperative pain control, supporting the use of thermo-hydrogels as a viable alternative to the standard of care in selected settings [23].

In knee osteoarthritis, HA in a hydrogel form combined with triamcinolone acetonide showed signals of efficacy and tolerability in a randomized feasibility study, with a faster onset of relief compared to hyaluronate alone. While not conclusive, this line of research indicates that viscoelastic gelling matrices can enhance the analgesic response of known active ingredients by modulating their intra-articular availability [29].

For topical/transdermal and mucosal indications, the use of muco-adhesive and bio-adhesives has improved drug contact time and adherence. The clinical literature on topical opioids in gel form shows heterogeneous results—with positive studies on the reduction in ulcerative pain and negative trials in specific dermatological settings—indicating that matrix composition, concentration, and the pain model strongly influence efficacy. This heterogeneity reinforces the need for pragmatic trials and shared core outcome sets for measures of pain, rescue consumption, and duration of effect [46].

Our non-DDS subgroup (excluded from the primary synthesis) confirms that the hydrogel matrix can modulate pain even without a drug through physical mechanisms (cooling, barrier, and lubrication), as shown by RCTs on hydrogel dressings in pediatric burns and the established practice of BCLs after PRK to reduce pain and promote re-epithelialization. The different hydrogel technologies identified in this review have distinct release and tissue–drug interaction profiles that help explain variability in analgesic outcomes. Thermo-responsive hydrogels (e.g., poloxamer-based) undergo a sol–gel transition at body temperature, allowing minimally invasive application as a liquid followed by in situ gelation. This property reduces the initial burst release, prolongs drug residence at the target site, and can provide a ‘single-shot’ alternative to continuous catheters. Hyaluronate-based hydrogels are viscoelastic and biointeractive; their high water content and affinity for extracellular matrix components can slow diffusion and enhance local retention of hydrophobic or charged molecules. Mucoadhesive or bioadhesive hydrogels increase contact time with mucosal surfaces, overcoming clearance mechanisms and improving local drug bioavailability. Together, these mechanisms shift the release profile from rapid diffusion to sustained, localized delivery, potentially reducing systemic exposure and extending analgesic effects. This context is relevant because, in combination with or as an alternative to DDS, hydrogels can offer multimodal strategies for local pain management [40,47].

A cross-cutting finding is the scarcity of clinical trials on hybrid hydrogel–nanotechnology systems and stimuli-responsive hydrogels in the analgesic field, despite a robust preclinical pipeline and evidence in other inflammatory areas. The lack of standardization (types of comparators, follow-up times, and patient-centered outcomes) and incomplete reporting of safety profiles limit the cumulativeness of the evidence. Recent advances have explored stimuli-responsive hydrogels for localized analgesic and antimicrobial drug delivery in the oral cavity and periodontitis. These “smart” systems respond to microenvironmental cues (e.g., local pH shifts, enzymatic activity, temperature changes, and ionic strength) to trigger drug release specifically at diseased sites (for instance, inflamed periodontal pockets) [48]. Future studies should (i) harmonize clinical endpoints (pain intensity, rescue consumption, and time to first rescue), (ii) integrate clinical pharmacokinetic–pharmacodynamic proxies (e.g., time to/duration of action for local anesthetics), (iii) compare hydrogel-based DDS with standards of care in adequately powered randomized trials, and (iv) implement safety registries for use in confined or high mechanical risk sites (where the non-DDS literature reports, albeit rarely, complications from migration into enclosed spaces) [2,49].

The main strength of this review is the breadth of the clinical mapping of hydrogel-based DDS for analgesics across various settings and administration routes, with classification by drug class and technology. Limitations arise from the heterogeneity of study designs, the variability of the matrices (often not fully characterized in the materials and methods), and the absence, in the mapped analgesic clinical trials, of widespread adoption of frontier hybrid platforms (e.g., liposomes-in-hydrogel, microparticles-in-gel). Bridging this clinical-translational gap—already well-delineated in engineering and drug delivery reviews—will require coordinated efforts between materials science, clinical pharmacology, and trial methodology [45,50].

For clinical practice, the data available to date suggest the following: (i) in perioperative settings, thermo-hydrogels with local anesthetics can be simplified alternatives to continuous catheter techniques in selected settings; (ii) in knee osteoarthritis, the addition of a corticosteroid to an HA hydrogel may accelerate analgesia compared to HA alone; and (iii) at topical/mucosal sites, mucoadhesion is a critical driver, but efficacy is sensitive to the pain model and formulation. These considerations must be verified with head-to-head comparative studies, standardized measures, and adequate follow-up to capture clinically significant differences and rare safety profiles [23,29].

A schematic diagram (Table 3) has been added to summarize the main findings of this review.

## 5. Conclusions

This scoping review highlights that hydrogel formulations used as DDS for analgesics show, across the different clinical settings mapped, convergent signals of benefit on pain outcomes and rescue medication consumption, with generally favorable tolerability profiles. This is especially true when the therapeutic goal is prolonged local exposure to analgesics with low systemic toxicity (e.g., in perioperative/incisional contexts, and via topical/mucosal routes). The consistency between the engineering rationale (e.g., in situ gelling, muco/bioadhesion, polymer network modulation, and stimuli-responsiveness) and the observed clinical endpoints supports the position of hydrogels as one of the most promising platforms for the targeted delivery of analgesics. Established technological trajectories—from thermo-hydrogels to stimuli-responsive systems—indeed converge towards more controlled and translatable release profiles, although they still suffer from formulation and reporting variability that hinders direct comparisons and formal meta-analyses.

A second key finding is that the “non-DDS ecosystem”—hydrogels used without a drug for cooling, barrier, or lubrication purposes—confirms the intrinsic ability of the matrix to modulate pain in a clinically appreciable manner (e.g., after PRK or in the early management of burns/wounds). This supports multimodal strategies where the hydrogel acts as a physical component complementary to the DDS. This landscape, while distinct from the primary scope, reinforces the idea that the material–tissue interface is an integral part of the efficacy perceived by the patient.

However, translational gaps remain: (i) a scarcity of clinical trials on hybrid hydrogel–nanotechnology platforms in analgesia despite encouraging preclinical signals; (ii) heterogeneity of comparators, follow-up, and patient-centered outcomes; and (iii) incomplete documentation of safety (rare local events, material–tissue interactions). The priorities for clinical research are, therefore, as follows: to standardize endpoints (pain intensity, time to/consumption of rescue medication, duration of effect, and QoL); to adopt head-to-head comparisons with the standard of care in adequately powered pragmatic RCTs; and to implement prospective safety registries for “constrained” or high mechanical risk sites. In parallel, the further maturation of responsive systems (pH, temperature, light/ultrasound) and injectable formulations could accelerate broader clinical adoption where local targeting is a critical determinant of the analgesic outcome.

Overall, the mapped clinical data indicate that hydrogels as DDS for analgesics constitute a field that is already useful in practice in specific settings and a growing platform for more sophisticated applications. Consolidating the evidence with rigorous comparative studies and harmonized reporting will be crucial to move from fragmented evidence to operational recommendations and, in the long term, to the safe and effective integration of “smart” technologies into pain management.

As a scoping review, this study relied on heterogeneous and often incompletely reported clinical data. Only descriptive statistics were applied, and no formal risk of bias or pooled effect estimates were conducted. These limitations restrict quantitative interpretation but provide a transparent overview of the current clinical landscape.

## Figures and Tables

**Figure 1 biomedicines-13-02465-f001:**
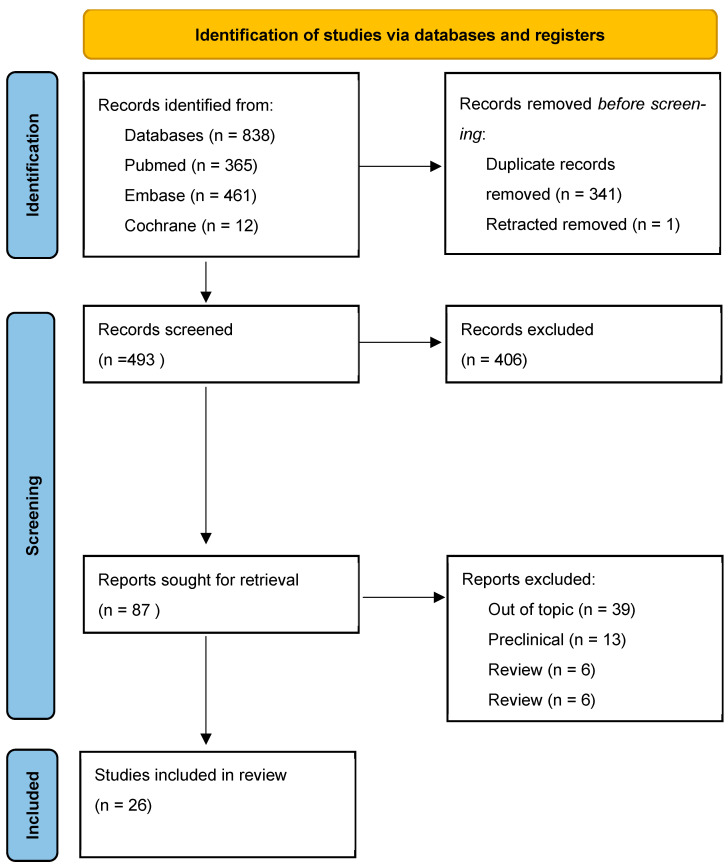
PRISMA flow diagram summarizing the study inclusion process.

**Figure 2 biomedicines-13-02465-f002:**
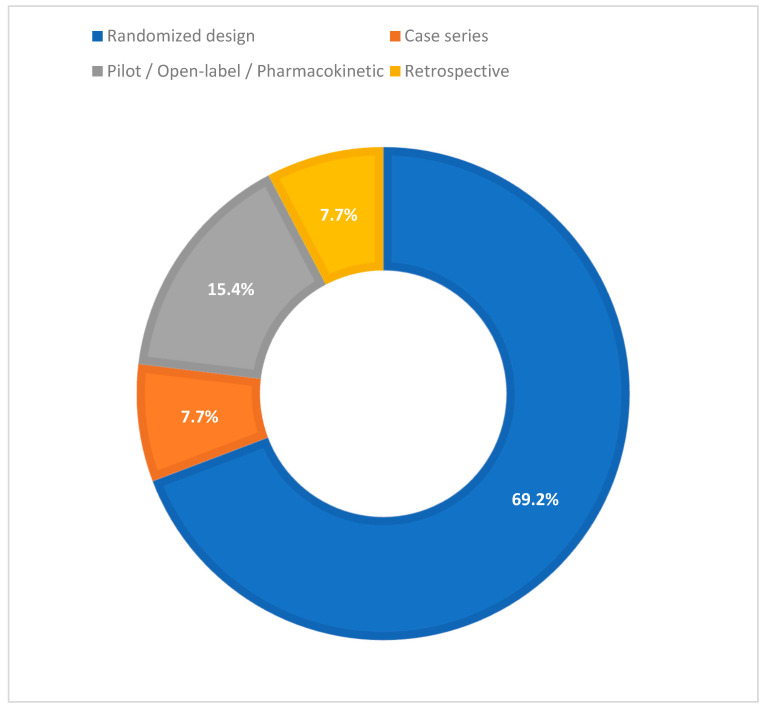
Distribution of study designs in the included clinical studies evaluating hydrogel formulations as drug delivery systems for analgesics.

**Figure 3 biomedicines-13-02465-f003:**
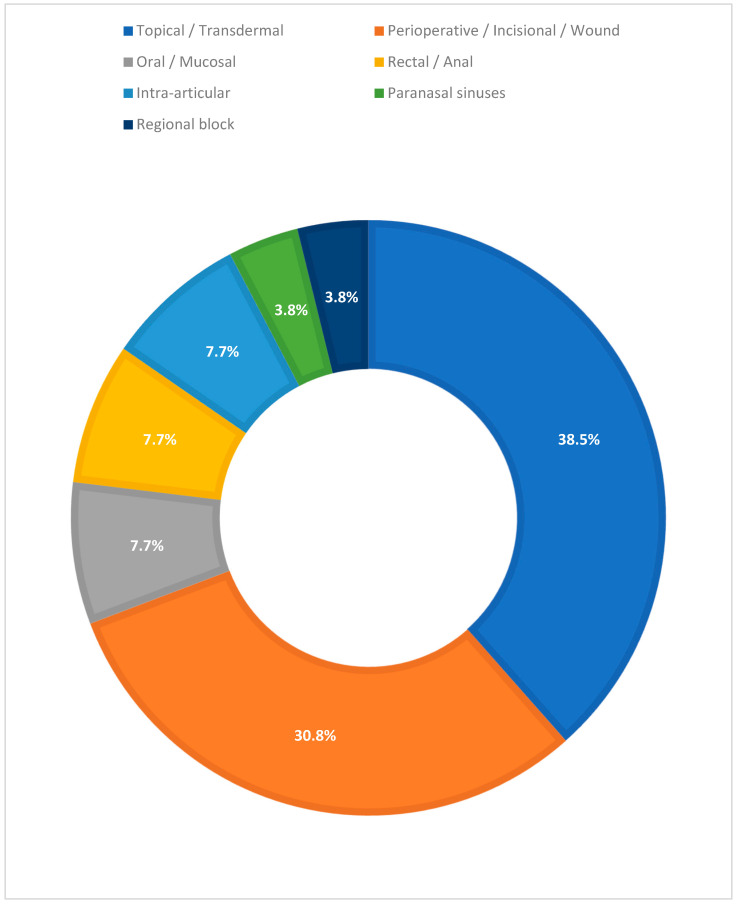
Distribution of application sites in the included clinical studies evaluating hydrogel formulations as drug delivery systems for analgesics.

**Figure 4 biomedicines-13-02465-f004:**
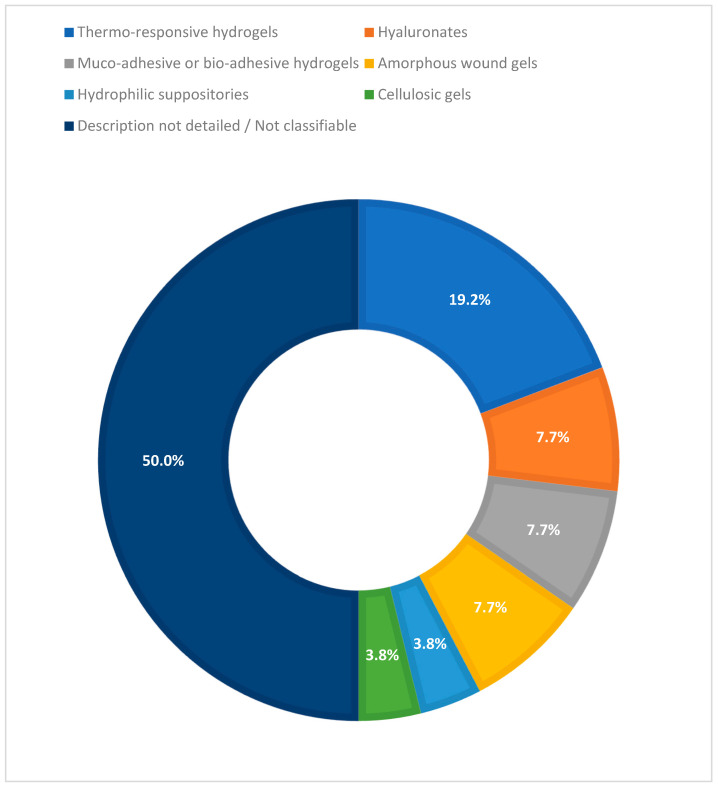
Distribution of hydrogel technologies used in the included clinical studies.

**Figure 5 biomedicines-13-02465-f005:**
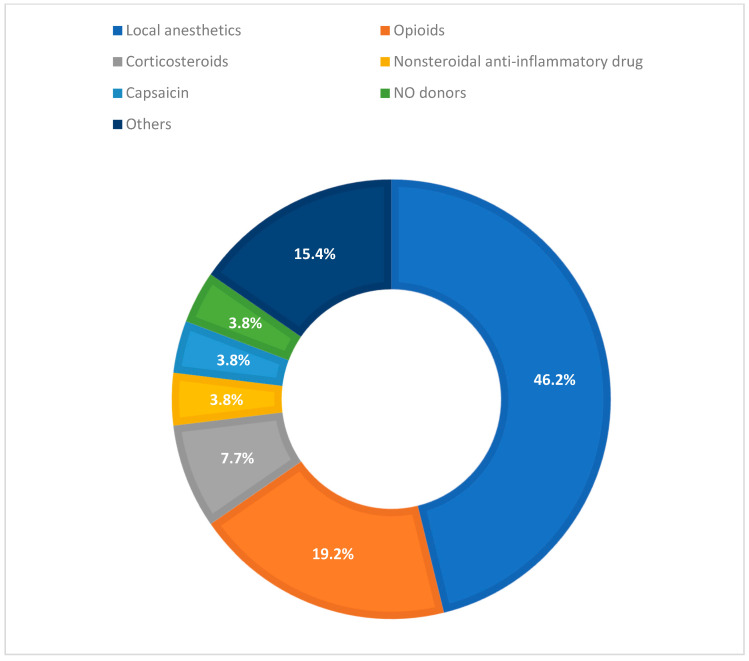
Distribution of drug classes in the included clinical studies.

**Table 1 biomedicines-13-02465-t001:** Drug Delivery Systems hydrogels characteristics and settings of usage.

Delivered Drugs	Number of Studies (RCTs)	Effect			Hydrogel Technology	References
		Improved	Neutral	Unclear		
Local anesthetics	12 (10)	10	1	1	Thermoresponsive, Other/Unspecified	[13,14,15,16,17,18,19,20,21,22,23,24]
Opioids	5 (1)	4	0	1	Amorphous wound gel, Unspecifed	[25,26,27,28]
Corticosteroids	2 (2)	2	0	0	Hyaluronic acid–based	[29,30]
Nitric Oxide	1 (1)	1	0	0	Topical nitrate ointment	[31]
NSAIDs	1 (1)	1	0	0	Unspecified	[32]
2-octyl cyanoacrylate, Aloe vera, Olea europea, Ozonized oil, Capsaicin	5 (1)	4	0	1	Other, Unspecified, Muco/Bioadhesive	[33,34,35,36,37]

**Table 2 biomedicines-13-02465-t002:** Non-Drug Delivery Systems hydrogels characteristics and settings of usage.

Hydrogel Group	Number of Studies (RCTs)	Role in Pain Control	Common Indications	Representative Hydrogels	Representative Citations
Ocular silicone hydrogels	12 (5)	Mechanical shield and lubrication of corneal epithelium after PRK; decreases nociceptive stimulation and blinking friction.	Acute ocular, Chronic ocular discomfort	Lotrafilcon A, Balafilcon A, Balafilcon A (silicone hydrogels)	[38,39]
Topical wound/burn hydrogels	12 (7)	Cooling/heat-sink, moist occlusive environment, and atraumatic removal reduce procedural and dressing-change pain.	Acute burn, Chronic wound	Burnaid, Oxyzyme, SockIt!, (hydrogel dressings)	[40,41]
Intradiscal hydrogel implants	6 (0)	Hydraulic cushioning and disk height support aim to reduce mechanical nociception; safety depends on implant stability.	Chronic disk pain, Chronic back	GelStix, Polyethylene glycol, Hyalodisc (hydrogel implants)	[42,43]
Other non-DDS hydrogels	8 (3)	Physical/biophysical modulation (barrier, lubrication, moisture) rather than pharmacological delivery.	Chronic hallux rigidus, Chronic knee Osteoarthritis	Polyvinyl alcohol hydrogel implant, hyaluronic acid hydrogel, Cross-linked sodium hyaluronate	[44]

**Table 3 biomedicines-13-02465-t003:** Schematic diagram summarizing the main findings of this scoping review.

Aspect	Key Findings	Notes
Hydrogel classification	DDS hydrogels (contain and release analgesic APIs) vs. non-DDS hydrogels (pain modulation via physical/biophysical mechanisms)	Non-DDS cataloged separately for transparency
Drug classes delivered	Local anesthetics (46%), opioids (19%), corticosteroids, NSAIDs, neuromodulators, nitric oxide donors	Predominantly perioperative, topical/mucosal, intra-articular
Hydrogel technologies	Thermo-responsive (poloxamer), hyaluronate-based, muco/bio-adhesive, amorphous wound gels, hydrophilic suppositories, cellulosic gels	No hybrid nanotechnology included in clinical studies
Administration sites	Topical/transdermal, perioperative/incisional, oral/mucosal, rectal/anal, intra-articular, sinus cavity, regional block	Broad distribution across acute and chronic pain settings
Analgesic effects	85% of DDS studies reported improved pain outcomes; reduced rescue medication; prolonged analgesic effect	More consistent benefits in acute/severe pain, variable in chronic/milder pain
Safety and tolerability	Generally favorable, but incomplete reporting of adverse events	Need for standardized safety registries
Non-DDS evidence	Physical pain modulation (cooling, barrier, lubrication) in burns, ocular, discogenic and mucosal conditions	Supports multimodal strategies combining matrix effects with pharmacological delivery

## Data Availability

Data are available on reasonable request from the corresponding author.

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
