# Peer review of "Hydrogel-Based Formulations to Deliver Analgesic Drugs: A Scoping Review of Applications and Efficacy"

_biomedicines, 2025, doi:10.3390/biomedicines13102465_

Round 1

Reviewer 1 Report

Comments and Suggestions for Authors

This scoping review is timely and addresses an important topic. The PRISMA-ScR approach is appropriate, and the coverage is broad. However, several points remain unclear and should be clarified before publication.

Major Questions

  1. Could the authors clarify more explicitly how DDS hydrogels (with drug delivery) were distinguished from non-DDS hydrogels (pain modulation via physical mechanisms)? Since non-DDS studies did not meet the inclusion criteria but were still catalogued, how should readers interpret this methodological decision? Please include the explanation into the content.
  2. Following, please address the explanation into the content. How did baseline severity of pain influence the reported outcomes? Were hydrogels equally effective in severe acute postoperative pain and in milder or chronic conditions?
  3. Could the authors expand on how different hydrogel types (e.g., thermo-responsive, hyaluronate, mucoadhesive) mechanistically modulate release kinetics and analgesic effects? For example, how do sol–gel transitions affect burst versus sustained release? Please add the explanation into the content.
  4. Please include stimuli-responsive hydrogel for localized analgesic drug delivery in dentistry and periodontitis in the research.
  5. What does the “**” symbol in Figure 1 mean? Could this be clarified either in the Methods section or directly in the figure legend?
  1. Would figure/table resolution (Figure 1) and abbreviation consistency be improved for clarity?
  2. Could terminology be harmonized (e.g., “thermo-responsive” vs. “temperature-responsive”)?
  3. Please create the schematic diagram for summarize your main finding before Conclusion and relate it link to content as final part of Discussion.
  4. Please include the statistical analysis technique used in the Method and apply with your data with more discussion.
  5. The limitation of this study should be stated in Conclusion.
  6. Please include “analgesic drug” in the Title.

Author Response

Comments 1: Could the authors clarify more explicitly how DDS hydrogels (with drug delivery) were distinguished from non-DDS hydrogels (pain modulation via physical mechanisms)? Since non-DDS studies did not meet the inclusion criteria but were still catalogued, how should readers interpret this methodological decision? Please include the explanation into the content.

Response 1: Thank you for point this out. We have clarified in the Methods (lines 123-135) how DDS and non-DDS hydrogels were distinguished and why the non-DDS studies were catalogued

Comments 2: Following, please address the explanation into the content. How did baseline severity of pain influence the reported outcomes? Were hydrogels equally effective in severe acute postoperative pain and in milder or chronic conditions?

Response 2: We agree with this comment. Therefore, we have added a statement in the Discussion (lines 302-311) clarifying how baseline pain severity influenced outcomes and the relative effectiveness in acute versus chronic conditions

Comments 3: Could the authors expand on how different hydrogel types (e.g., thermo-responsive, hyaluronate, mucoadhesive) mechanistically modulate release kinetics and analgesic effects? For example, how do sol–gel transitions affect burst versus sustained release? Please add the explanation into the content.

Response 3: We have expanded the Discussion (lines 341–353) to explain how different hydrogel types influence release kinetics and analgesic effects, including the role of sol–gel transitions in reducing burst release and enabling sustained delivery

Comments 4: Please include stimuli-responsive hydrogel for localized analgesic drug delivery in dentistry and periodontitis in the research.

Response 4: We have added a section in the Discussion (around lines 360-365) introducing stimuli-responsive hydrogels

Comments 5: What does the “**” symbol in Figure 1 mean? Could this be clarified either in the Methods section or directly in the figure legend?

Response 5: We have removed the ‘**’ symbol from Figure 1 as it was a remnant of a previous template and had no meaning

Comments 6: Would figure/table resolution (Figure 1) and abbreviation consistency be improved for clarity?

Response 6: We have improved the resolution of Figure 1 and ensured consistent use of abbreviations throughout the manuscript for greater clarity.

Comments 7: Could terminology be harmonized (e.g., “thermo-responsive” vs. “temperature-responsive”)?

Response 7: We have harmonized terminology across the manuscript, using ‘thermo-responsive’ consistently instead of mixed terms such as ‘temperature-responsive

Comments 8: Please create the schematic diagram for summarize your main finding before Conclusion and relate it link to content as final part of Discussion.

Response 8: We have added a schematic diagram before the Conclusion summarizing the main findings and linked it to the final part of the Discussion.

Comments 9: Please include the statistical analysis technique used in the Method and apply with your data with more discussion.

Response 9: We have added in the Methods (lines 145-148) that only descriptive statistics were used, as appropriate for a scoping review

Comments 10: The limitation of this study should be stated in Conclusion.

Response 10: We have added a statement on the study’s limitations at the end of the Conclusion section

Comments 11: Please include “analgesic drug” in the Title.

Response 11: We have revised the title to include the term ‘analgesic drug’

Reviewer 2 Report

Comments and Suggestions for Authors
  1. the title looks general, while in the abstract the authors focuse on  Hydrogel-based DDS for analgesics. I recommend to change to title according to this point. (line 98 confirms this)
  2. Introduction looks good, well-arranged. in the last paragraph the authors must highlight the main purpose of this study.
  3. Material and method: well-explained
  4. line148-181: it will be better to present those percentages in graph too.the same for section 3.1
  5. line260-262: Reference?
  6. result section needs more figures to highlight the differences?
  7. Discussion looks well arranged.

Author Response

Comments 1: the title looks general, while in the abstract the authors focuse on  Hydrogel-based DDS for analgesics. I recommend to change to title according to this point. (line 98 confirms this)

Response 1: Thank you for point this out. We have revised the title

Comments 2: Introduction looks good, well-arranged. in the last paragraph the authors must highlight the main purpose of this study.

Response 2: We agree with this comment. Therefore, we have added a sentence at the end of the Introduction (around lines 102–106) to explicitly highlight the main purpose of the study

Comments 3: Material and method: well-explained

Response 3: Thank you for pointing this out.

Comments 4: line148-181: it will be better to present those percentages in graph too. the same for section 3.1

Response 4: We added graphs in the sections.

Comments 5: line260-262: Reference?

Response 5: We have added reference to support the statement

Comments 6: result section needs more figures to highlight the differences?

Response 6: thank you, there are four figures explaining the results.

Comments 7: Discussion looks well arranged.

Response 7: Thank you for your appreciation

Round 2

Reviewer 1 Report

Comments and Suggestions for Authors

The authors improve the content proproly according to the comments.

Author Response

Comments 1: The authors improve the content proproly according to the comments.
Response 1: Thank you for your appreciation.

Reviewer 2 Report

Comments and Suggestions for Authors
  1. title: we do not have something like analgestic drug delivery. the title should be like:

Hydrogel-based Formulations to deliver analgesic drugs: A Scoping Review of Applications and Efficacy

Author Response

Comments 1: Hydrogel-based Formulations to deliver analgesic drugs: A Scoping Review of Applications and Efficacy

Response 1: We have changed the title according to the suggestions.